

# The phylogeny of desmostylians revisited: proposal of new clades based on robust phylogenetic hypotheses

Kumiko Matsui[1,2] and Takanobu Tsuihiji[3,4]

[1] The Kyushu University Museum, Kyushu University, Fukuoka, Japan
[2] The University Museum, The University of Tokyo, The University of Tokyo, Tokyo, Japan
[3] Department of Earth and Planetary Science, Graduate school of Science, The University of Tokyo, Tokyo, Japan
[4] Department of Geology and Paleontology, National Museum of Nature and Science, Tsukuba, Japan

## ABSTRACT

**Background**. Desmostylia is a clade of extinct aquatic mammals with no living members. Today, this clade is considered belonging to either Afrotheria or Perissodactyla. In the currently-accepted taxonomic scheme, Desmostylia includes two families, 10 to 12 genera, and 13–14 species. There have been relatively few phylogenetic analyses published on desmostylian interrelationship compared to other vertebrate taxa, and two main, alternative phylogenetic hypotheses have been proposed in previous studies. One major problem with those previous studies is that the numbers of characters and OTUs were small.

**Methods**. In this study, we analyzed the phylogenetic interrelationship of Desmostylia based on a new data matrix that includes larger numbers of characters and taxa than in any previous studies. The new data matrix was compiled mainly based on data matrices of previous studies and included three outgroups and 13 desmostylian ingroup taxa. Analyses were carried out using five kinds of parsimonious methods.

**Results**. Strict consensus trees of the most parsimonious topologies obtained in all analyses supported the monophyly of Desmostylidae and paraphyly of traditional Paleoparadoxiidae. Based on these results, we propose phylogenetic definitions of the clades Desmostylidae and Paleoparadoxiidae based on common ancestry.

## INTRODUCTION

Desmostylia is a clade of extinct aquatic mammals with no living members (*Repenning, 1965*; *Inuzuka, 1984*; *Inuzuka, 2000b*; *Inuzuka, 2000c*; *Domning, 2002*; *Gingerich, 2005*). The phylogenetic affinities of the clade among mammals are still debated, having been hypothesized as belonging to Afrotheria (*Domning, Ray & McKenna, 1986*), Perissodactyla (*Cooper et al., 2014*; *Rose et al., 2014*) or Paenungulatomorpha (*Gheerbrant, Filippo & Schmitt, 2016*), due to their specialized morphology (Fig. 1).

In the currently-accepted taxonomic scheme, Desmostylia includes two families, 10 to 12 genera, and 13–14 species (*Shikama, 1966*; *Kohno, 2000*; *Inuzuka, 2005*; *Domning & Barnes, 2007*; *Barnes, 2013*; *Beatty & Cockburn, 2015*; *Chiba et al., 2016*). The two

Corresponding author
Kumiko Matsui,
kumiko_matsui@me.com

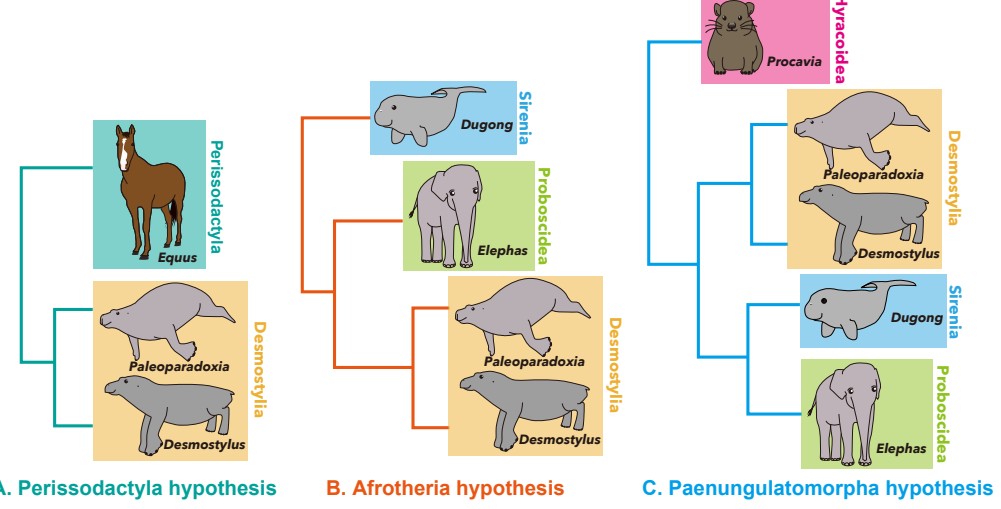

A. Perissodactyla hypothesis    B. Afrotheria hypothesis    C. Paenungulatomorpha hypothesis

**Figure 1** **Summary of hypotheses on phylogenetic affinities of Desmostylia within Mammalia.** (A) Perissodactyla hypothesis, (B) Afrotheria hypothesis, (C) Paenungulatamorpha hypothesis.

families are Desmostylidae *Osborn, 1905*, and Paleoparadoxiidae *Reinhart, 1959*. Presently, Desmostylidae includes *Ashoroa laticosta*, *Cornwallius sookensis*, *Ounalashkastylus tomidai*, *Kronokotherium brevimaxillare*, *Desmostylus japonicus*, *D. hesperus* and *D. (Vanderhoofius) coalingensis* (*Domning & Barnes, 2007*; *Inuzuka, 2005*; *Chiba et al., 2016*). Paleoparadoxiidae has been considered to include two subfamilies, Behemotopsinae including *Seuku emlongi*, *Behemotops proteus* and *Behemotops katsuiei* (*Domning, Ray & McKenna, 1986*; *Inuzuka, 2000b*; *Beatty & Cockburn, 2015*) and Paleoparadoxiinae that includes *Archaeoparadoxia weltoni*, *Paleoparadoxia tabatai*, *Neoparadoxia repenningi* and *Neoparadoxia cecilialina* (*Barnes, 2013*). It is noteworthy, however, that results of some phylogenetic analyses do not support this taxonomic scheme (e.g., *Beatty & Cockburn, 2015*).

## Previous studies on desmostylian phylogenetic interrelationships

There have been relatively few phylogenetic analyses published on desmostylian interrelationships compared to other vertebrate taxa. The results of previous studies are summarized here (Fig. 2). *Domning, Ray & McKenna (1986)* performed the first phylogenetic analysis that included Desmostylia. Before their study, *Osborn (1905)* and *Reinhart (1953)* suggested that Desmostylia is closely related to Sirenia and Proboscidea, and this hypothesis had been widely accepted. However, it had not been demonstrated to which of these two clades Desmostylia is more closely related. *Domning, Ray & McKenna (1986)* analyzed phylogenetic relationships among *Prorastomus*, *Protosiren*, crown Sirenia, the primitive tethytherian *Minchenella*, *Anthracobune*, *Moeritherium*, *Barytherium*, *Prodeinotherium*, *Deinotherium*, *Paleomastodon*, crown Proboscidea and Desmostylia including *Behemotops proteus*, *B. emlongi*, *Paleoparadoxia*, *Cornwallius* and *Desmostylus*. As a result, Desmostylia was found to be most closely related to Proboscidea. In addition,

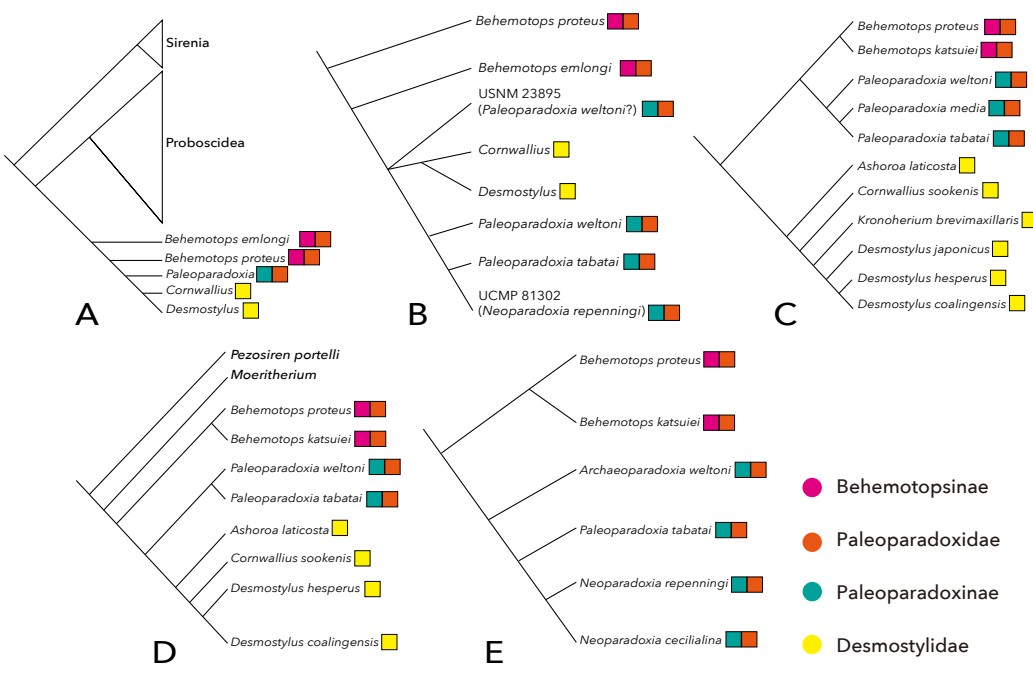

**Figure 2** **Previously-proposed hypotheses on the phylogenetic interrelationship of Desmostylia.** (A) the topology of *Domning, Ray & McKenna (1986)* (credit) Smithsonian Libraries, (B) the topology of *Clark (1991)*, (C) the topology of *Inuzuka (2005)*, (D) the topology of *Beatty (2009)*, (E) the topology of *Barnes (2013)* (credit) Natural History Museum Los Angeles, Pink: Behemotopsinae, Orange: Paleoparadoxiidae, Green: Paleoparadoxiinae, Yellow: Desmostylidae.

*Domning, Ray & McKenna (1986)* proposed the hypothesis that *Minchenella* was a suitable candidate for the ancestor (or the sister taxon) of the clade consisting of Desmostylia and Proboscidea, suggesting the origin of the latter two clades in Asia.

*Clark (1991)* performed the first phylogenetic analysis of desmostylian interrelationships including the new species of *Paleoparadoxia* that he described. His analysis included *Behemotops emlongi*, *B. proteus*, *Cornwallius*, *Desmostylus*, *Paleoparadoxia tabatai*, *P. weltoni* and two undescribed desmostylian specimens as OTUs. The result corroborated the monophyly of *Paleoparadoxia* and strongly supported a clade consisting of *Desmostylus*, *Cornwallius* and *Paleoparadoxia*. However, the relationship between *Paleoparadoxia* and the clade including *Desmostylus* and *Cornwallius* was unresolved.

*Inuzuka* (*2000b*, *Inuzuka, 2005*) proposed a new phylogenetic tree of Desmostylia encompassing all valid desmostylian species including new primitive desmostylid materials described in *Inuzuka (2000b)*. His data matrix includes more post-cranial characters than were used in previous phylogenetic analyses of desmostylians. However, the methods employed for these phylogenetic analyses were not described in either paper. According to Inuzuka's results, Desmostylia consists of two clades, Desmostylidae (*A. laticosta*, *C. sookensis*, *K. brevimaxillare*, *D. hesperus*, *D. japonicus* and *D. coalingensis*) and Paleoparadoxiidae (*B. proteus*, *B. katsuiei*, *P. weltoni*, "*P. media*" and "*P. tabatai*").

*Beatty (2009)* assembled a new matrix based on previous studies and included new data on *Cornwallis sookensis*. He used *Moeritherium* and *Pezosiren portelli* as outgroups of Desmostylia and included nearly all species of Desmostylia. The tree that *Beatty (2009)* obtained is different in topology from the one in *Inuzuka (2000b*, *Inuzuka, 2005)* in that *Behemotops* spp. were placed below the node containing other traditional paleoparadoxiids, making the traditionally-recognized family Paleoparadoxiidae paraphyletic.

*Barnes (2013)* made a new data matrix for analyzing the phylogenetic position of a new paleoparadoxiid as well as the interrelationships of Paleoparadoxiidae. His data matrix includes numerous post-cranial skeletal characters. In the cladogram that he obtained, three formerly-known species of *Paleoparadoxia* (separated into three genera by *Barnes, 2013*) formed the clade Paleoparadoxiinae. The problem with his analysis, however, is that it was based on the assumption of the traditional Paleoparadoxiidae including *Behemotops* being monophyletic. This assumption was not rigorously tested and had been challenged by *Beatty (2009)*.

A more recent analysis by *Chiba et al. (2016)* was based on a data matrix modified from *Beatty (2009)*. *Chiba et al. (2016)* added two molar characters to *Beatty (2009)*'s matrix and analyzed the phylogenetic position of *Ounalashkastylus*. Their topology is ((*Moeritherium, Pezosiren, Anthracobnidae), ((Behemotops proteus, B. katsuiei), (Archaeoparadoxia weltoni, Paleoparadoxia tabatai, (Ashoroa laticosta, (Cornwallius sookensis, (Ounalashkastylus tomidai, (Desmostylus hesperus, Vanderhoofius coalingensis*, cf. *Vanderhoofius* sp.))))))). This tree has a topology similar to the one obtained in *Beatty (2009)*, with *Ounalashkastylus* placed between *Cornwallius* and the clade consisting of *Desmostylus* and *Vanderhoofius* spp.

## Purpose of this study

The above review of past phylogenetic analyses points to problems with these studies. Firstly, not all valid desmostylian species were included in most previous analyses. Secondly, almost all analyses were based on the assumption that Desmostylia is a member of Afrotheria. Recently, however, this assumption was challenged based on phylogenetic analyses indicating that Desmostylia is a part of Perissodactyla (*Cooper et al., 2014*; *Rose et al., 2014*) or Paenungulatomorpha (*Gheerbrant, Filippo & Schmitt, 2016*). If this is the case, using afrotherians (e.g., proboscideans and/or sirenians) as outgroups for a phylogenetic analysis of desmostylian interrelationships is problematic. It is therefore necessary to run phylogenetic analyses using alternative outgroups representing different hypotheses of affinities of Desmostylia to examine effects of outgroup selections. Thirdly, for the numbers of taxa being analyzed, relatively few characters were used in past analyses. To summarize, global phylogeny of Desmostylia still needs to be analyzed by (1) incorporating all currently-accepted species, (2) using several outgroups reflecting various hypotheses of desmostyian affinities and (3) producing a data matrix with more characters.

In order to rectify these three problems, a new, largest data matrix for desmostylian interrelationships was assembled in this study and was analyzed using different outgroups reflecting currently-proposed hypotheses of desmostylian affinities. The resulting trees were then used to obtain a robust topology independent of outgroups in order to propose new phylogenetic definitions of the clades Desmostylidae and Paleoparadoxiidae.

## MATERIALS & METHODS

### Taxon sampling

#### Outgroups

In this study, three separate analyses were performed using different outgroups to account for uncertainty of desmostylian affinities with other mammals. Desmostylia has been hypothesized as belonging to Afrotheria, Perissodactyla or Paenungulatomorpha. In the case of the Afrotherian hypothesis, it is also not certain whether Desmostylia is closer to Sirenia or Proboscidea. Herein the following three analyses using different sets of outgroups were conducted. These analyses cover all appropriate outgroups suggested by the three phylogenetic hypotheses above.

(1) Analysis 1. *Anthracobune* spp. as the outgroup (coding based on *Cooper et al. (2014)*),

(2) Analysis 2. *Pezosiren portelli*, a primitive sirenian, and *Moeritherium* spp., a primitive proboscidean, as the outgroups (coding based on NMNS PV-20726, 20970–4, Andrews (*1904* and *1906*), *Holroyd et al. (1996)*, and *Delmer et al. (2006)*),

(3) Analysis 3. *Anthracobune* spp., *Pezosiren portelli* and *Moeritherium* spp. as the outgroups.

#### In-group taxa

In this study, 13 species of desmostylians were included as OTUs. All presently-accepted desmostylian species were included. A possible exception is *Kronokotherium brevimaxillare* which has been considered a junior synonym of *Desmostylus hesperus* (*Domning, 1996*) and is known only from highly fragmentary specimens (*Pronina, 1957*; *Beatty, 2009*). The following is the list of OTUs with sources for coding.

1. *Behemotops proteus* (based on USNM 244035; *Domning, Ray & McKenna, 1986*; *Beatty & Cockburn, 2015*; *Ray, Domning & McKenna, 1994*).
2. *Behemotops katsuiei* (based on AMP 22; *Inuzuka, 2000b*; *Inuzuka, 2009*).
3. *Seuku emlongi* (based on USNM 244033 and 186889; *Domning, Ray & McKenna, 1986*; *Beatty & Cockburn, 2015*; *Ray, Domning & McKenna, 1994*).
4. *Archaeoparadoxia weltoni* (based on UCMP 114285; *Clark, 1991*).
5. *Paleoparadoxia tabatai* (based on NMNS PV-5601; *Shikama, 1966*; *Ijiri & Kamei, 1961*).
6. *Neoparadoxia repenningi* (based on UCMP 81302; *Inuzuka, 2005*).
7. *Neoparadoxia cecilialina* (based on LACM 150150; *Barnes, 2013*).
8. *Ashoroa laticosta* (based on AMP 21; *Inuzuka, 2000b*; *Inuzuka, 2011*).
9. *Cornwallius sookensis* (based on USNM 11073, 11075, 181738, 181740, 181741, and 214740; *Beatty, 2006*; *Beatty, 2009*).
10. *Ounalashkastylus tomidai* (based on *Chiba et al., 2016*; *Jacobs et al., 2007*; *Jacobs et al., 2009*).
11. *Desmostylus japonicus* (based on NMNS PV-5600; GSJ-F02071; *Kohno, 2000*; *Yoshiwara & Iwasaki, 1902*).
12. *Desmostylus hesperus* (based on UHR-18466; GSJ-F7743; UCMP 32742; *Ijiri & Kamei, 1961*; *Inuzuka, 1980a*; *Inuzuka, 1980b*; *Inuzuka, 1981b*; *Inuzuka, 1981a*; *Inuzuka, 1982*; *Inuzuka, 1988*; *Inuzuka, 2009*).

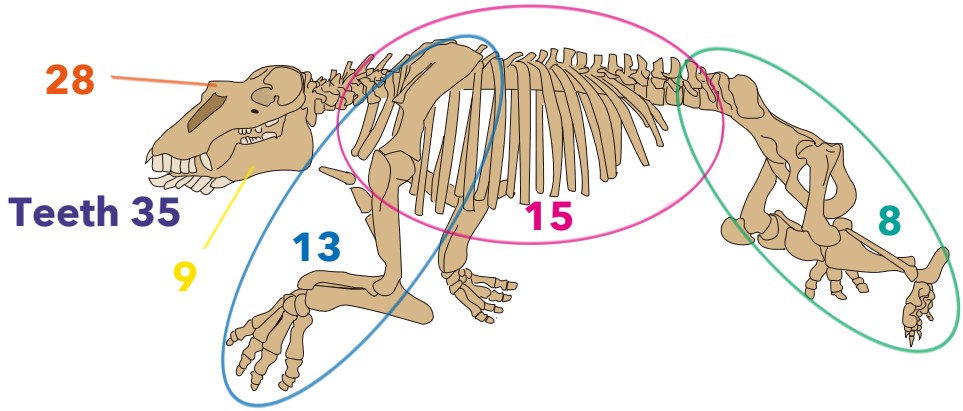

**Figure 3** **Distribution of morphological characters by the region used in the new matrix assembled in the present study.** The skeleton is *Neoparadoxia repenningi* modified from *Panofsky (1998)* and its figure by Pete Nuding, courtesy of the SLAC National Accelerator Laboratory. Purple: teeth, Orange: skull, Yellow: mandible, Blue: forelimb, Pink: trunk, Green: hindlimb.

13. *Desmostylus* (*Vanderhoofius*) *coalingensis* (based on USNM 244489; UCMP 39990; *Reinhart, 1959*; *Inuzuka, 2005*; *Beatty, 2009*).

## Software and analysis

The data matrix was assembled in Mesquite v 3.6 (*Maddison & Maddison, 2011*). Analyses were conducted with equally weighted parsimony with PAUP* (*Swofford, 2002*) version 4.0a, build 165 for Macintosh using the heuristic search algorithm with Tree Bisection Reconnection (TBR) branch swapping (saving 10 trees per replication). Branch support was estimated with bootstrap resampling method (10,000 replicates). Phylogenetic trees were illustrated by using the geoscalePhylo function in the strap package (*Gradstein, Ogg & Schmitz, 2012*) for the statistical programming language R (*R Core Team, 2017*). The divergence time estimation was also calculated by geoscalePhylo function in trap package.

## Characters and data matrices

Firstly, analyses were run based on previously-published character matrices (*Inuzuka, 2000b*; *Barnes, 2013*; *Chiba et al., 2016*; *Clark, 1991*; *Beatty, 2009*) to verify the published tree topologies. Secondly, those matrices were compiled, with coding revised and new characters added. Overall, 110 morphological characters were employed in the new matrix (Fig. 3). Character descriptions and data matrices are provided in File S1 and Table S1.

## RESULTS

### Reproducibility of previous data matrices

Among previously-published data matrices, only the data matrix of *Inuzuka (2005)* did not produce the original topology presented in the paper (Fig. S1).

### Analyses based on a new data matrix

All results are shown in Fig. 4 and Fig. S2–S3, S5–S6, and S8-S9. Bootstrap consensus trees obtained in all the analyses showed the identical topology (Fig. 4; Figs. S3, S6, S9) whereas

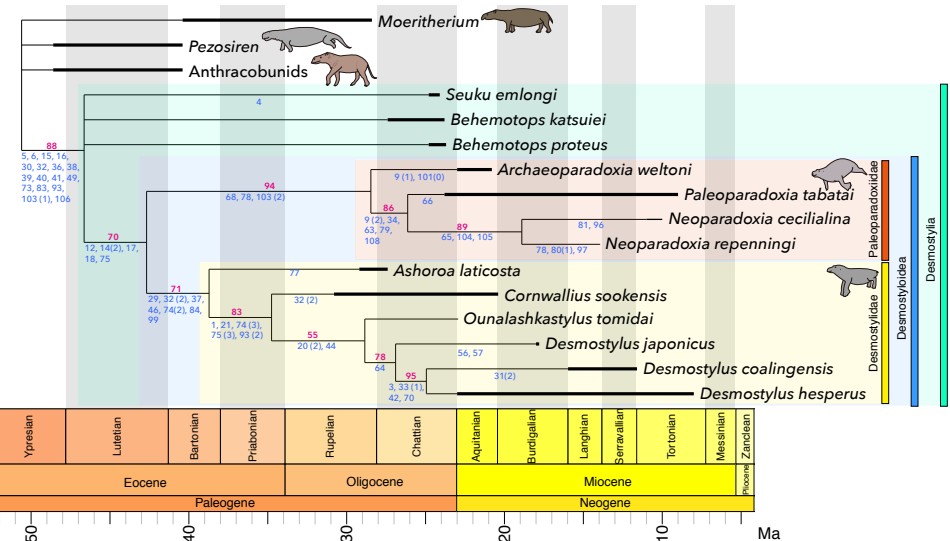

**Figure 4** **Time-calibrated strict consensus tree resulting from the present analyses.** The number written in red below each node represents the bootstrap value (in %). The numbers written in blue indicate characters and character states representing synapomorphies for each node. Black bar: geological range, Green: Desmostylia, Blue: Desmostyloidea, Orange: Paleoparadoxiidae, Yellow: Desmostylidae. L = 205, CI = 0.668, RI = 0.682, RC = 0.456, HI = 0.332, G-fit = − 78.950.

strict consensus trees (Figs. S2, S5, S8) of these analyses had partly different topologies. However, all these topologies (Fig. 4, and S2–S3, S5–S6, S8–S9) agree on both traditional Paleoparadoxiinae including *Archaeoparadoxia*, *Paleoparadoxia* and *Neoparadoxia* and traditional Desmostylidae including *Ashoroa*, *Cornwallius*, *Ounalashkastylus* and *Desmostylus* being monophyletic as well as on Desmostylidae + Paleoparadoxiinae forming a clade. On the other hand, Paleoparadoxiidae *sensu Inuzuka (2000b*, 2005) and *Barnes (2013)* that includes *Paleoparadoxia*, *Archaeoparadoxia*, *Neoparadoxia*, *Seuku* and *Behemotops* spp. was not recovered as a clade. The positions of *Behemotops* spp. and *Seuku* differs among the strict consensus trees obtained in Analyses 1–3. In all the bootstrap consensus trees of these analyses, *Behemotops* and *Seuku* formed an unresolved polytomy with the clade containing the remaining desmostylians. These genera thus diverged before the split between Paleoparadoxiinae and Desmostylidae.

# DISCUSSION

## Reproducibility of data matrices

The analysis based on *Inuzuka*'s (*2005*) original data matrix produced a completely unresolved polytomy with no resolution. This matrix includes a relatively few characters for the number of OTUs, likely contributing to non-resolution of the tree topology.

## Characters supporting each clade in the present analyses

Although not all character distribution patterns were shared among the strict consensus trees of Analyses 1 through 3 (Figs. S4, S7, S10), many common synapomorphies were

found for major clades. Such synapomorphies identified in all the strict consensus trees are described below.

The monophyly of traditional Desmostylidae consisting of *Ashoroa*, *Cornwallius*, *Ounalashkastylus* and *Desmostylus* was supported by the presence of 7 or more cusps on M3 (c. 29(1)), conical and tusk-like lower incisors (c. 32( 2)), no passage anterior to the external auditory meatus connecting to the skull roof (c. 37(1)), presence of an anterior orbital groove (c. 46(1)), having cancellous bones of vertebrae (centrum) (c. 75(3)) and shallow and wide shape of intertubercular groove in humeus(c. 93(2)). The monophyly of traditional Paleoparadoxiinae consisting of *Archaeoparadoxia*, *Paleoparadoxia* and *Neoparadoxia* was supported by the mandibular symphysis rotated anteroventrally (c. 68(1)), 14 or 15 theoretic vertebrae (c. 78(1)), and a flat femoral shaft (c. 103 (2)). The clade consisting of Paleoparadoxiinae + Desmostylidae was supported by the absence of the p3 paraconid (c. 12(1)), fused double roots of p3 and p4 (c. 14(2)), swollen and appressed molar cusps (c. 17(1)), enlarged P4-M3 hypoconulid and entoconid (c. 18(1)) and having osteosclerosis bones of vertebrae (centrum) (c. 75(1)). Synapomorphies of Desmostylia are a tusk root enlarged in diameter (c. 5(1)), an enlarged lower canine (c. 6(1)), the hypoconid and entoconid reduced in height in p4 talonid (c. 15(1)) a transversely broad hypoconulid shelf of m3 (c. 16(1)), transversely aligned lower incisors (c. 30(1)), a flattened or conical and tusk-like lower incisor (c. 32(1 & 2)), absent of foramen within squamosal passing anterior from external auditory meatus (c. 36(1)), elongating to much behind alveolus of incisors and canine in posterior part of premaxilla (c. 38(1)), high and closed ventrally external auditory meatus (c. 39(1)), elongated paraoccipital process (c. 40(1)), the presence of the foramen post-zygomaticus (c. 41(1)), basioccipital bone's length less than half of the width of the foramen magnum (c. 49(1)), paired sternebrae (c. 73(1)), exist of the ring like shape epiphyseal line in centrum (c. 83(1)), shallow and narrow intertubercular groove in humerus (c. 93(1)), distal surface inclined medially in capitate bone (c. 103(1)), and tibia is medially twisted with its distal articular surface facing laterally (c. 106(1)). The monophyly of *Desmostylus* (*D. japonicus* + *D. hesperus* + "*Vanderhoofius*" *coalingensis*) was supported by the sigmoid upper margin of mandibular body (c. 64(1)). The monophyly of *D. hesperus* + "*Vanderhoofius*" *coalingensis* was supported by the loss of the upper canine (c. 3(1)), the presence of one pair of upper incisors (c. 33(1)), premaxilla contacting the frontal (c. 42(1)) and the laterally convex interalveolar margin in the diastema of the mandible (c. 70(1)). The monophyly of *Neoparadoxia* was supported by a small angle between the anterior and posterior margins of the coronoid process (c. 65(1)), the tibia–fibula articulation enlarged and extended proximally (c. 104(1)) and the astragalar facet on the tibia tilted at least 60 degrees from horizontal (c. 105(1)).

## Comparisons with MPTs and synapomorphies for clades obtained in previous studies

In this study, a new data matrix was constructed including more characters and taxa than those used in previous studies. The present MPT topologies are clearly different from the one presented in *Inuzuka (2000b*, 2005) but are mostly consistent with the one in *Beatty (2009)*. An assumption by *Barnes (2013)* that both Paleoparadoxiinae and

Paleoparadoxiidae were monophyletic was rejected herein. In addition, the relationship among *Archaeoparadoxia*, *Paleoparadoxia* and Desmostylidae was unresolved in *Chiba et al. (2016)* likely because their matrix did not include enough characters. In this study, the data matrix consisting of more characters successfully resolved the relationship among these three taxa.

The synapomorphies identified in the present study are somewhat different from those proposed by previous studies. *Clark (1991)* identified two synapomorphies for traditional Paleoparadoxiinae and three synapomorphies for Desmostylidae + Paleoparadoxiinae. However, the present analyses did not find any of these characters diagnosing these clades except for *Clark*'s (*1991*) Character 29 (Character 68 in the present data matrix). As an OTU, *Clark*'s (*1991*) matrix included an undescribed specimen (USNM 23895) not included in the present analyses, possibly causing differences in synapomorphies of these clades.

*Inuzuka (2005)*, on the other hand, identified four synapomorphies for Desmostylia, six for Desmostylidae, three for traditional Paleoparadoxiinae and two for *Desmostylus*. None of those synapomorphies identified in *Inuzuka* (*2005*; his Characters 1, 3, 8, 12, 14, 15, 31, 32, 34 and 35) supported these clades in the present analyses. The strict consensus topologies obtained in the present analyses are different from the one presented in *Inuzuka (2005)*. Therefore, such differences may be expected.

## TAXONOMY OF DESMOSTYLIA

The present results suggest that the previously-proposed taxa Desmostylidae and Paleoparadoxiinae are monophyletic and valid. On the other hand, Paleoparadoxiidae including *Behemotops* (*Inuzuka, 2000c*; *Barnes, 2013*; *Inuzuka, 2009*) turned out to be paraphyletic. Therefore, the currently-used taxon Paleoparadoxiidae needs to be re-defined as a clade excluding *Behemotops*, leaving it with the same taxonomic content as the currently-used Paleoparadoxiinae (*Beatty, 2009*). *Behemotops* and *Seuku* are not included in either monophyletic Desmostylidae or Paleoparadoxiidae. Additionally, *Vanderhoofius* (= "*Desmostylus*") *coalingensis*, *D. hesperus* and *D. japonicus* formed a clade in the strict consensus trees of all present analyses. Therefore, these results support the hypothesis of *Kohno (2000)* and *Santos & Parham (2016)* that *Vanderhoofius* is a junior synonym of *Desmostylus*.

### New definition of desmostylian clades

In this study, the monophyly of traditional Paleoparadoxiidae was rejected. Desmostylian families have been defined based on a traditional convention of simply enumerating included taxa. Such an approach was regarded as non-evolutionary by de Queiroz and Gauthier (*1990*, *1992*, and *1994*). These authors instead proposed phylogenetic definitions of taxon names, i.e., defining taxon names in terms of common ancestry, which has resulted in the proposal of the formal International Code of Phylogenetic Nomenclature (PhyloCode) governing the naming of clades (*Cantino & De Queiroz, 2010*). Their rationale is followed here and traditional desmostylian family names are converted to clade names with new definitions following the PhyloCode rules.

DESMOSTYLIDAE OSBORN 1905 (CONVERTED CLADE NAME)

Definition: Desmostylidae refers to the clade consisting of *Desmostylus hesperus* Marsh 1888 and all organisms or species that share a more recent common ancestor with *D. hesperus* than with *Paleoparadoxia tabatai* Tokunaga 1939.

Comments: Because the Order Desmostylia is currently divided into two families, Desmostylidae and Paleoparadoxiidae, it is appropriate to convert these taxa to branch- or stem-based clades so that all desmostylian species except for a few, early-diverging forms (e.g., those regarded as Family indeterminate by *Beatty & Cockburn (2015)* are included in one of these clades. All taxa traditionally regarded as constituting Desmostylidae formed a clade in the present analyses (Fig. 4). Therefore, the converted clade of Desmostylidae include the same set of currently valid taxa as the traditional Family Desmostylidae.

Based on the current analyses, Desmostylidae is diagnosed by the following characteristics: the presence of seven or more cusps on M3 (c. 29(1)), conical and tusk-like lower incisors (c. 32( 2)), no passage anterior to the external auditory meatus connecting to the skull roof (c. 37(1)), presence of an anterior orbital groove (c. 46(1)), having cancellous bones of vertebrae (centrum) (c. 75(3)) and shallow and wide shape of intertubercular groove in humweus(c. 93(2)).

PALEOPARADOXIIDAE *Reinhart, 1959* (CONVERTED CLADE NAME)

Definition: Paleoparadoxiidae refers to the clade consisting of *Paleoparadoxia tabatai* Tokunaga 1939 and all organisms or species that share a more recent common ancestor with *P. tabatai* than with *Desmostylus hesperus* Marsh 1888.

Comments: Traditionally-recognized paleoparadoxiids formed a paraphyletic group and thus did not form a clade in all present analyses (Fig. 4), necessitating a revision of the content of the taxon. Based on the present analyses, the clade Paleoparadoxiidae is diagnosed by the following synapomorphies: the mandibular symphysis rotated anteroventrally (c. 68(1)), 14 or 15 theoretic vertebrae (c. 78(1)), and a flat femoral shaft (c. 103 (2)).

DESMOSTYLOIDEA *Osborn, 1905* (CONVERTED CLADE NAME)

Definition: Desmostyloidea refers to the clade originating with the most recent common ancestor of *Desmostylus hesperus* Marsh 1888 and *Paleoparadoxia tabatai* Tokunaga 1939.

Comments: The new clade Desmostyloidea includes Desmostylidae and Paleoparadoxiidae as its subclades. Because these two clades are defined above as branch-based clades, any member of Desmostyloidea belongs to either Desmostylidae or Paleoparadoxiidae.

The following synapomorphies of Desmostyloidea were identified in the present analyses: The clade consisting of Paleoparadoxiinae + Desmostylidae was supported by the absence of the p3 paraconid (c. 12(1)), fused double roots of p3 and p4 (c. 14(2)), swollen and appressed molar cusps (c. 17(1)), enlarged P4-M3 hypoconulid and entoconid (c. 18(1)) and having osteosclerosis bones of vertebrae (centrum) (c. 75(1)).

DESMOSTYLIA *Reinhart, 1953* (CONVERTED CLADE NAME)

Definition: Desmostylia refers to the clade originating with the first organism or species to possess as an apomorphy the transversely broad hypoconulid shelf of the third molar, as inherited by *Desmostylus hesperus* Marsh 1888.

Comments: The order Desmostylia was established by *Reinhart, 1953* for the genera *Desmostylus* and *Cornwallius*. Since then, several new genera have been referred to this order by *Reinhart (1959)*, *Domning, Ray & McKenna (1986)*, *Inuzuka (2000a)*, *Barnes (2013)*, *Beatty & Cockburn (2015)* and *Chiba et al. (2016)*. In the present analyses, such genera were all found to be included in one clade and share numerous synapomorphies.

Several alternative phylogenetic definitions of Desmostylia are possible, but the newly defined clade should approximate traditional use of the name. The node-based definition would be "the clade originating with the most recent common ancestor of *Desmostylus hesperus* Marsh 1888, *Paleoparadoxia tabatai* Tokunaga 1939, *Seuku emlongi* (*Domning, Ray & McKenna, 1986*), *Behemotops proteus* (*Domning, Ray & McKenna, 1986*) and *Behemotops katsuiei Inuzuka, 2000a*." This definition, however, would exclude from the clade earlier-diverging or "stem" species on this lineage. The branch-based definition, on the other hand, would be "the clade consisting of *Desmostylus hesperus* Marsh 1888 and all organisms or species that share a more recent common ancestor with *D. hesperus* than with *Anthracobune pinfoldi Pilgrim, 1940*, *Trichechus manatus Linnaeus, 1758*, or *Elephas maximus Linnaeus, 1758*", considering currently hypothesized sister clades of Desmostylia. However, the exact relationships of Desmostylia with other mammalian clades are still debated and it is possible that other clades will turn out to be more closely related to Desmostylia than those that have been hypothesized. Considering that such a case would result in a wildly different taxonomic content of Desmostylia than that currently recognized, this branch-based definition also appears inappropriate.

Desmostylia was originally proposed by *Reinhart (1953)* for *Osborn*'s (*1905*) Desmostylidae and *Hay*'s (*1924*) Desmostyliformes. It included currently-recognized *Paleparadoxia*, *Cornwallius*, and *Desmostylus*. Although various recent studies identified diagnostic features of Desmostylia (e.g., *Inuzuka, 2005*; *Matsui, 2017*; *Matsui et al., 2018*), they did not attempt define the name of the clade Desmostylia. In this study we newly established the apomorphy-based definition for this clade. The clade defined in this way includes not only the derived clades Paleoparadoxiidae and Desmostylidae that share the "bundled, pillar-like" teeth, but also earlier-diverging members *Seuku* and *Behemotops* possessing a transversely broad hypoconulid shelf that would have been a precursor of the highly specialized dental morphology of those clades. Considering that members of Desmostylia have been recognized based on such unique dental morphology present in derived species, it is most reasonable to adopt the apomorphy-based definition based on a dental characteristic as proposed here.

## CONCLUSIONS

In this study, a new data matrix was assembled for analyzing phylogenetic interrelationships of Desmostylia. The results of the analyses support a monophyletic Paleoparadoxiinae consisting of *Archaeoparadoxia*, *Paleoparadoxia* and *Neoparadoxia* as well as a Desmostylidae consisting of *Ashoroa*, *Cornwallius*, *Ounalashkastylus*, and *Desmostylus*. In addition, *Behemotops* and *Seuku* turned out to form an unresolved polytomy with the clade of Paleoparadoxiinae + Desmostylidae. Based on these results, the phylogenetic

definitions of Desmostylia, Desmostylidae and Paleoparadoxiidae, as well as a new clade Desmostyloidea, are proposed.

**Institutional abbreviations**

| | |
|---|---|
| **AMP** | Ashoro Museum of Paleontology, Hokkaido, Japan |
| **GSJ** | Geological Survey of Japan, Ibaraki, Japan |
| **LACM** | Natural History Museum of Los Angeles County, Los Angeles, California, USA |
| **NMNS** | National Museum of Nature and Science, Tokyo, Japan |
| **UCMP** | University of California Museum of Paleontology, Berkeley, California, USA |
| **UHR** | Hokkaido University Museum, Sapporo, Japan |
| **USNM** | Department of Paleobiology, US National Museum of Natural History, Smithsonian Institution, Washington, D.C., USA. |

# ACKNOWLEDGEMENTS

Thanks are due to Nicholas Pyenson, David Bohaska (USNM), Mark Goodwin, and Patricia Holroyd (UCMP), Jorge Velez-Juarbe, Samuel A. McLeod, and Vanessa R. Rhue (LACM), Naoki Kohno (NMNS), and CH Tsai (NMNS, currently National Taiwan University), Hiroshi Sawamura, Tatsuro Ando, and Tatsuya Shinmura (AMP), Yoshitsugu Kobayashi, Tomonori Tanaka, Tsogtbaatar Chinzorig, Kota Kubo (UHM) for allowing us to study desmostylian specimens under their care. KM also thanks Kazuyoshi Endo, Takenori Sasaki, Makoto Manabe, and Naoki Kohno for helpful discussions on her Ph.D. dissertation including a chapter on which this paper is based. We are grateful for the constructive reviews by the editor, John Hutchinson, and reviewers, Brian L. Beatty and Daryl P. Domning.

## Funding

Kumiko Matsui received support from the Japan Society for the Promotion of Science Research grant for Young Scientists (JSPS 16J00546) and the Sasakawa Scientific Research Grant 2018-6028 from the Japan Science Society. The funders had no role in study design, data collection and analysis, decision to publish, or preparation of the manuscript.

## Grant Disclosures

The following grant information was disclosed by the authors:
Japan Society for the Promotion of Science Research grant for Young Scientists: JSPS 16J00546.
Sasakawa Scientific Research Grant 2018-6028 from the Japan Science Society.

## Competing Interests

The authors declare there are no competing interests.

## Author Contributions

- Kumiko Matsui conceived and designed the experiments, analyzed the data, prepared figures and/or tables, authored or reviewed drafts of the paper, approved the final draft.
- Takanobu Tsuihiji conceived and designed the experiments, analyzed the data, authored or reviewed drafts of the paper, approved the final draft.

## Data Availability

All raw data are available in the Supplemental Files.

Specimens are at the Ahoro Museum of Paleontology, Ashoro, Hokkaido, Japan (Behemotops katsuiei AMP 22, Ashoroa laticosta: AMP 21), the Geological Survey of Japan, Ibaraki, Japan (Desmostylus hesperus: GSJ-F7743, Desmostylus japonicus: GSJ-F02071), Natural History Museum of Los Angeles County, Los Angeles,California, USA (Neoparadoxia cecilialina: 150150), National Museum of Nature and Science, Tokyo, Japan (Desmostylus japonicus: NMNS PV-5600, Paleoparadoxia tabatai: NMNS PV-5601, Pezosiren portelli: NMNS PV-20726, 20970–4), the University of California Museum of Paleontology, Berkeley, California, USA (Archaeoparadoxia weltoni: UCMP 114285, Neoparadoxia repeninngi: UCMP 81302, Desmostylus hesperus: UCMP 32742, Desmostylus (Vanderhoofius) coalingensis: UCMP 39990), the Hokkaido University Museum, Sapporo, Japan (Desmostylus hesperus: UHR-18466), and the National Museum of Natural History, Smithsonian Institution, USA (Behemotops proteus USNM 244035; Cornwallius sookensis USNM 11073, 11075, 181738, 181740, 181741, and 214740; Desmostylus (Vanderhoofius) coalingensis USNM 244489)

## Supplemental Information

Supplemental information for this article can be found online at http://dx.doi.org/10.7717/peerj.7430#supplemental-information.

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
