# Peer review of "The phylogeny of desmostylians revisited: proposal of new clades based on robust phylogenetic hypotheses"

_PeerJ, doi:10.7717/peerj.7430_

## Round 0.1 · original submission · Minor Revisions

The 2 reviewers available have some fairly minor critiques for the paper to be revised with. Please do clarify the character state "size" and other qualitative definitions, to make the work more reproducible, as per Reviewer 1's comments. And as usual please include a full point-by-point Response to all comments. Thank you and sorry the the delay with this review round! There should be no further review required if your revisions are thorough.

·

Basic reporting

No comment.

Experimental design

No comment.

Validity of the findings

No comment.

Additional comments

This study is a welcome and well-executed reevaluation of desmostylian relationships and interrelationships.It usefully incorporates the character sets used by earlier workers, and advances our understanding of the group. It is also well-written, with only very minor corrections of the English needed.

I have made numerous editorial corrections and suggestions on the marked-up copy of the MS. that is attached here, which is to be shared with the authors. These include questions about the definitions of some of the characters used, and some matters of nomenclature (such as the authorship of "Desmostyloidea"). In general, I am not a fan of cladistic classification, or unranked classifications; however, the authors are certainly at liberty to use these procedures.

My most substantive criticism of this MS. is that many of the character-state definitions need more work: specifically, expressions like "smaller than" vs. "much smaller than"; "short" vs. "elongate"; etc. The authors should imagine themselves in the position of an investigator becoming acquainted with desmostylians for the first time, and trying to score a new specimen using definitions like these. They should reconsider all the definitions in the Character List, and expand on such definitions using ratios or other quantifications, positions of a feature relative to other features of a bone, or even illustrations to clarify the distinctions between character states.

·

Basic reporting

The usage of English throughout is fairly consistent, though there are several minor typos that should be correctly to improve clarity. These appear to be errors that might have persisted because of errors in the Word dictionary, as they are repeated throughout and often are proper nouns. They include:
Vonderhoofius - should be Vanderhoofius
canie - should be canine
prpteus - should be proteus (this is found in Figure 2)
emlongis - should be emlongi (this is found in Figure 2)
repeninngi - should be repenningi (this is found in Figure 2)

The authors are thorough in their citation of appropriate references. The structure of the article is excellent, data shared, and the results to hypotheses make sense.

Experimental design

The authors have defined a clear question and have done a thorough job of analysis. This is by far the most thorough analysis this group has received, and the outcomes make sense. They have gone above and beyond to make things replicable and test the replicability of odler studies as well.

Validity of the findings

The results make sense and conclusions are the rational outcome of the results. They are fairly consistent with regard to avoiding speculation. This includes their conservative inclusion of D. coalingensis, which other authors have dismissed as a nomen dubium without much clear support. Though I have my doubts about D. coalingensis as a valid taxon, they explain that ascertaining this is not in the scope of the paper and thus do not attempt to argue to their exclusion. This was wise.

Additional comments

I am impressed by the care and detail by which this study was conducted. It doesn't overreach the results and comes to a rational conclusion based on results alone.
I only mark this as minor revisions being needed for the corrections to the typos. Scientific content is sound.

---

## Round 0.2 · accepted · Accept

Thank you for your careful revisions and patient attention to the reviews. I have checked the changes and am satisfied- so congratulations, your paper is accepted!